# STALENESS-BASED SUBGRAPH SAMPLING FOR LARGE-SCALE GNNS TRAINING

## ABSTRACT

Training Graph Neural Networks (GNNs) on large-scale graphs is challenging. The main difficulty is to obtain accurate node embeddings while avoiding the neighbor explosion problem. Many of the existing solutions use historical embeddings to tackle this challenge. Specifically, by using historical embeddings for the out-of-batch nodes, these methods can approximate full-batch training without dropping any input data while keeping constant GPU memory consumption. However, it still remains nascent to specifically design a subgraph sampling method that can benefit these historical embedding-based methods. In this paper, we first analyze the approximation error of node embeddings caused by using historical embeddings for out-of-batch neighbors and prove that this approximation error can be minimized by minimizing the staleness of historical embeddings of out-of-batch nodes. Based on the theoretical analysis, we design a simple yet effective Staleness score-based Subgraph Sampling method (S3) to benefit these historical embedding-based methods. The key idea is to first define the edge weight as the sum of the staleness scores of the source and target nodes and then apply graph partitioning to minimize edge cuts, with each resulting partition as a mini-batch during training. In this way, we can explicitly minimize the approximation error of node embeddings. Furthermore, to deal with the dynamic changes of staleness scores during training and improve the efficiency of graph partitioning, we design a fast algorithm to generate mini-batches via a local refinement heuristic. Experimental results show that (1) our S3 sampling method can further improve historical embedding-based methods and set the new state-of-the-art, and (2) our fast algorithm is 3x faster than re-partitioning graph from scratch on the large-scale ogbn-products dataset with 2M nodes. In addition, the consistent improvements on all three historical embedding-based methods (GAS, GraphFM, and LMC) also show the generalizability of our subgraph sampling method.

## 1 INTRODUCTION

Graph neural networks (GNNs) (Kipf & Welling, 2017; Velickovic et al., 2018; Xu et al., 2018; Gasteiger et al., 2018; Corso et al., 2020; Chen et al., 2020) are powerful methods to learn both node and graph representations for various downstream tasks, such as node property prediction (Hu et al., 2020; 2021), link property prediction (Zhang & Chen, 2018), and graph property prediction (Gilmer et al., 2017; Wu et al., 2018; Yang et al., 2019). However, the scalability of GNNs is often limited due to neighbor explosion which makes the size of the node receptive field increase exponentially with respect to the number of layers/hops. For example, given a graph with $k$ neighbors per node and a GNN with $\ell$ layers, an astounding $k^\ell$ nodes are required to learn the embedding for a single node which is computationally costly. This problem hinders the applications of GNNs to large-scale graphs, such as citation networks and social networks.

Existing works aiming to improve the scalability of the GNNs and the efficiency of GNN training mainly focus on designing advanced sampling strategies. Specifically, node-wise sampling methods (Hamilton et al., 2017; Ying et al., 2018; Huang et al., 2023) recursively sample a fixed number of neighbors for each node, while layer-wise sampling methods (Chen et al., 2018; Huang et al., 2018; Zou et al., 2019) sample nodes in each layer independently, leading to a constant number of nodes in each layer. Instead of sampling nodes or edges across GNN layers, subgraph sampling methods (Zeng et al., 2019; 2021; Chiang et al., 2019) first sample subgraphs as mini-batches and

then build a full GNN on the subgraphs. Given a sampled mini-batch $\mathcal{B}$, the embedding $h_u^{\ell+1}$ for a target node $u$ is obtained by aggregating information from sampled neighbors, as

$$h_u^{\ell+1} = f_\theta^{\ell+1}(h_u^\ell, \{h_v^\ell\}_{v \in \mathcal{N}(u) \cap \mathcal{B}})$$

where $h_u^\ell$ is the node embedding of node $u$ at layer $l$ and $\mathcal{N}(u)$ denotes 1-hop neighbors of node $u$. In this case, the information from unsampled neighbors is discarded during this training step. To mitigate this problem, several historical embedding-based methods (Fey et al., 2021; Yu et al., 2022; Shi et al., 2023) have been proposed to use historical embeddings of the unsampled neighbors (i.e., out-of-batch neighbors) as an affordable approximation. The idea of historical embeddings is originally introduced in (Chen et al., 2017) and (Cong et al., 2020). GAS (Fey et al., 2021) provides a reliable framework to use historical embeddings and sets the state-of-the-art in various benchmarks. As shown in Fey et al. (2021), the message passing with historical embeddings can be formulated as

$$
\begin{aligned}
h_u^{\ell+1} &= f_\theta^{\ell+1}(h_u^\ell, \{h_v^\ell\}_{v \in \mathcal{N}(u)}) \\
&= f_\theta^{\ell+1}(h_u^\ell, \{h_v^\ell\}_{\underbrace{v \in \mathcal{N}(u) \cap \mathcal{B}}_{\text{in-batch neighbors}}} \cup \{h_v^\ell\}_{\underbrace{v \in \mathcal{N}(u) \setminus \mathcal{B}}_{\text{out-of-batch neighbors}}}) \\
&\approx f_\theta^{\ell+1}(h_u^\ell, \{h_v^\ell\}_{v \in \mathcal{N}(u) \cap \mathcal{B}} \cup \underbrace{\{\bar{h}_v^\ell\}_{v \in \mathcal{N}(u) \setminus \mathcal{B}}}_{\text{historical embeddings}})
\end{aligned}
\tag{1}
$$

where $\bar{h}_v^\ell$ is the corresponding historical embedding of node $v$ at layer $\ell$. GAS shows that using historical embeddings can lead to constant GPU memory consumption with respect to input node size without dropping any data. GraphFM (Yu et al., 2022) and LMC (Shi et al., 2023) further enhance GAS by reducing the staleness of historical embeddings from the algorithmic perspective and improving the accuracy of gradient estimation, respectively. Detailed illustrations of GAS and GraphFM are provided in Figure 3 in Appendix A.

However, there are several challenges related to the historical embedding-based methods. Intuitively, a higher quality of the historical embeddings of out-of-batch nodes may translate to better final node embeddings. However, it is not theoretically clear how the quality of historical embeddings affects the quality of output node embeddings. Second, existing historical embedding-based methods such as GAS (Fey et al., 2021), GraphFM (Yu et al., 2022), and LMC (Shi et al., 2023), directly apply Cluster-GCN (Chiang et al., 2019) to construct subgraph mini-batches, whereas there is no dedicated method to generate mini-batches during training that can guarantee the quality of historical embeddings of out-of-batch neighbors. Third, since the historical embeddings dynamically change during training, it is worth considering generating subgraph mini-batches on-the-fly, which can be costly. How to re-sample subgraphs efficiently is another key challenge.

In this paper, we provide our solution to systematically overcome these challenges and further improve the historical embedding-based methods. Our contributions can be summarized in three folds.

- We theoretically analyze the impact of the quality of historical embeddings on the quality of output node embeddings. Specifically, we first leverage the path-based view (Gasteiger et al., 2022) of GNNs with ReLU activations to decompose the aggregations among hidden representations of in-batch nodes and historical embeddings of out-of-batch nodes. This enables us to compute the approximation error of the learned node embeddings, which can be upper bounded in terms of the staleness of the historical embeddings.

- We design a novel subgraph sampling method, called S3, based on the staleness scores such that the historical embeddings of out-of-batch neighbors do not bring much information loss compared to their up-to-date hidden representation counterparts. More specifically, we first define the edge weight as the sum of the staleness scores of the source and target nodes. Then we apply graph partitioning to generate mini-batches. In this way, we can explicitly minimize the approximation error of learned node embeddings.

- To deal with the dynamic changes of staleness scores during training and improve the efficiency of graph partitioning, we further design a fast algorithm to avoid re-partitioning the graph from scratch. Specifically, at the $t$-th training epoch, instead of directly partitioning the graph $G^t$ into $G_1^t, G_2^t, ..., G_M^t$, we perform refinement on the partitioning results $G_1^{t-1}, G_2^{t-1}, ..., G_M^{t-1}$ at the previous epoch via a local refinement heuristic (Karypis & Kumar, 1998b). Our refinement algorithm is 3x faster than graph partitioning from scratch on the large-scale ogbn-products dataset with 2M nodes.

The analysis on the approximation error of node embeddings in Section 3 and experimental results in Section 4 show that our S3 sampling method can further improve the existing historical embedding-based methods both theoretically and empirically and set the new state-of-the-art. The consistent improvements on all three historical embedding-based methods (GAS, GraphFM, and LMC) also demonstrate the generalizability of our proposed subgraph sampling method.

# 2 BACKGROUND AND RELATED WORK

Due to the extensive number of nodes in large-scale graphs, mini-batch training is crucial for efficient training. To enable mini-batch training on large-scale graphs, two main categories of approaches have been proposed, namely sampling methods and pre-computing methods. In this section, we first summarize different sampling methods and introduce the historical embedding-based methods in Section 2.1. Note that our method aligns with this research direction. Next, we summarize pre-computing methods and other techniques for large-scale GNNs training in Section 2.2.

## 2.1 SAMPLING-BASED METHODS

A main challenge for large-scale GNNs training is the neighbor explosion problem (Hamilton et al., 2017). Specifically, for each target node that we want to get an output from a GNN, the number of involved nodes in the computation graph grows exponentially with the number of GNN layers. Different sampling methods have been proposed to tackle this problem.

**Node-wise sampling.** Instead of considering all neighbors, GraphSAGE (Hamilton et al., 2017) and PinSAGE (Ying et al., 2018) sample a fixed number of neighbors for each node. Specifically, GraphSAGE uniformly samples neighbors, while PinSAGE adapts based on node importance. Following GraphSAGE, VR-GCN (Chen et al., 2017) further reduces the number of sampled neighbors by integrating historical embeddings to reduce the estimator variance. GraphFM-IB (Yu et al., 2022) applies feature momentum to reduce the number of sampled neighbors while maintaining good performance. Note that although these methods can reduce the memory cost by sampling only part of the neighbors for each node, they still suffer from the neighbor explosion problem since the number of incorporated nodes still grows exponentially with the number of GNN layers.

**Layer-wise sampling.** This type of methods samples a fixed number of nodes in each layer, and hence the number of nodes only grows linearly with the number of layers. For example, FastGCN (Chen et al., 2018) samples nodes independently for each layer. Huang et al. (2018) and Zou et al. (2019) further consider between-layer correlations to improve accuracy. However, these methods may incur computational overhead due to the expensive sampling algorithm.

**Subgraph sampling.** Instead of sampling nodes or edges across GNN layers, subgraph sampling methods first sample subgraphs and then build a full GNN on the subgraphs. Therefore, it resolves the neighbor explosion problem since the number of nodes involved in the computation graph is the same as the subgraph size. For example, GraphSAINT (Zeng et al., 2019) provides several light-weight and efficient samplers, including random node sampler, random edge sampler, and random walk-based sampler, to construct subgraphs. SHADOW (Zeng et al., 2021) and IBMB (Gasteiger et al., 2022) sample subgraphs based on PageRank scores. Cluster-GCN (Chiang et al., 2019) samples dense subgraphs by minimizing inter-partition edges using graph partitioning algorithms. The subgraph sampling method is also related to distributed training on large graphs (Chen et al., 2023; Peng et al., 2022; Wan et al., 2022; Bai et al., 2023).

**Scalable GNNs with historical embeddings.** Building on top of sampling methods, some methods further use historical embeddings as an affordable approximation to improve the scalability and model performance. The main advantage of using historical embeddings is that we can approximate full-batch training without dropping any data while keeping constant GPU memory consumption. The idea of historical embeddings is originally introduced in (Chen et al., 2017) and (Cong et al., 2020). GAS (Fey et al., 2021) provides a reliable framework and codebase to use historical embeddings and sets the state-of-the-art in various benchmarks. GraphFM-OB (Yu et al., 2022) and LMC (Shi et al., 2023) further improve GAS by using feature momentum and considering gradient estimation respectively. However, how to construct mini-batches $\mathcal{B}$ has not been well studied. Here, we design a new sampling method to construct mini-batches, which can further improve the historical embedding-based methods.

## 2.2 PRE-COMPUTING METHODS AND OTHER TECHNIQUES

**Pre-processing methods.** On the other hand, pre-processing methods (Wu et al., 2019; Frasca et al., 2020; Yu et al., 2020; Sun et al., 2021; Zhang et al., 2022) first compute informative features for each node and then feed these features into subsequent models. In this case, the nodes can be treated as independent samples, and we can easily pass the nodes in mini-batches. This technique offers an alternative perspective for tackling the challenges of large-scale GNNs training.

**Other techniques.** Our method is orthogonal to many advanced techniques (Shi et al., 2020; Huang et al., 2020; Sun et al., 2021; Chien et al., 2021; Kong et al., 2022; Zhang et al., 2021; Han et al., 2023; Li et al., 2021; Duan et al., 2022) for large-scale GNNs training. For example, Shi et al. (2020) uses label propagation to boost model performance. MLPInit (Han et al., 2023) accelerates the training of GNNs by initializing the weights of GNNs with the weight of their converged PeerMLP. Graph transformer (Ying et al., 2021; Rampášek et al., 2022) has the ability to leverage all node information. However, the dense attention computations make it less efficient. Improving the efficiency (Poli et al., 2023) and enabling mini-batch training (Chen et al., 2022) of graph transformers on large-scale graphs is also an important research direction.

## 3 METHOD

Historical embeddings has the unique advantage of enabling approximation of full-batch training without dropping any data while keeping constant GPU memory consumption (Fey et al., 2021). However, the main challenge is that there is no sampling method specifically designed for these historical embedding-based methods. For example, GAS, GraphFM, and LMC (Fey et al., 2021; Yu et al., 2022; Shi et al., 2023) simply apply Cluster-GCN (Chiang et al., 2019) to generate mini-batches practically. In Cluster-GCN, the unweighted graph partitioning algorithm (*e.g.* METIS) (Karypis & Kumar, 1998a; Dhillon et al., 2007) is directly used to convert the input graph into several subgraphs such that the number of inter-partition edges is minimized. However, there is lack of justification on whether Cluster-GCN can generate informative mini-batches to harness historical embeddings. Indeed, in this section, we show that it is not optimal. Specifically, in Section 3.1, we first analyze the approximation error of the learned node embeddings caused by using historical embeddings for out-of-batch nodes. We prove that the approximation error can be minimized by minimizing the staleness of historical embeddings. We then present our staleness score-based sampling in Section 3.2.

### 3.1 APPROXIMATION ERROR ANALYSIS

As shown in Equation 1, the main idea of GNNs with historical embeddings is to approximate full-batch embedding $h_u^{\ell+1}$ for each node $u$ by aggregating embeddings $\{h_v^\ell\}_{v \in \mathcal{N}(u) \cap \mathcal{B}}$ for in-batch neighbors and historical embeddings $\{\bar{h}_v^\ell\}_{v \in \mathcal{N}(u) \setminus \mathcal{B}}$ for out-of-batch neighbors. However, there exists an approximation error if $\bar{h}_v^\ell \neq h_v^\ell$ for $v \in \mathcal{N}(u) \setminus \mathcal{B}$. In this section, we analyze how historical embeddings affect the approximation error of the final node embedding $\tilde{h}_u^L$, which motivates the design of our new sampling method.

**Theorem 1.** *Given a GNN with a linear, graph-dependent aggregation and ReLU activations, the embedding approximation error, i.e., the error between the full-neighborhood propagation embedding $h_u^L$ and the actual aggregated embedding $\tilde{h}_u^L$ by using historical embeddings,*

$$\|h_u^L - \tilde{h}_u^L\|$$

*can be minimized by minimizing*

$$\sum_{v \in \mathcal{N}(u) \setminus \mathcal{B}} \sum_{\ell=1}^{L-1} C_v^\ell \|h_v^\ell - \bar{h}_v^\ell\|.$$

Here $\sum_{\ell=1}^{L-1} C_v^\ell \|h_v^\ell - \bar{h}_v^\ell\|$ represents the overall quality of the historical embeddings at all $L-1$ layers where $C_v^\ell$ is a weight that depends on both graph structure and model parameters, and we want to minimize the sum of these terms of all out-of-batch neighbors. The proof for Theorem 1 is provided in Appendix B. Intuitively, based on Equation 1, we can see that the approximation error depends on both the number of out-of-batch neighbors and the quality of the historical embeddings of these out-of-batch neighbors. However, previous methods such as GAS, GraphFM, and LMC

only consider the number of out-of-batch nodes and use unweighted graph partitioning algorithms to minimize it. Unfortunately, they do not directly consider the quality of historical embedding which limits their performance.

## 3.2 STALENESS SCORE-BASED SUBGRAPH SAMPLING

**Staleness scores of historical embeddings.** As discussed in Section 3.1 and Theorem 1, the embedding approximation error can be minimized by minimizing the weighted sum of $\|h_v^\ell - \bar{h}_v^\ell\|$ for all out-of-batch neighbors at all $\ell = 1, \cdots, L-1$ layers. Here $\|h_v^\ell - \bar{h}_v^\ell\|$ is defined as the *staleness score* of the historical embedding of node $v$ at layer $\ell$ (Fey et al., 2021; Yu et al., 2022), which measures the Euclidean distance between full-neighborhood propagation embedding $h_v^\ell$ and historical embedding $\bar{h}_v^\ell$. Formally, for each node $v$, the staleness score $s_v^\ell$ at layer $\ell$ is

$$s_v^\ell = \|h_v^\ell - \bar{h}_v^\ell\|. \tag{2}$$

**Optimization objective.** Based on Theorem 1, to sample a mini-batch $\mathcal{B}$, our optimization objective is

$$\arg\min_{\mathcal{B}} \sum_{u \in \mathcal{B}} \sum_{v \in \mathcal{N}(u) \setminus \mathcal{B}} \sum_\ell C_v^\ell s_v^\ell. \tag{3}$$

And we want to minimize the weighted sum of the staleness scores for all out-of-batch neighbors.

In subgraph sampling, we want to convert the input graph $G = (\mathcal{V}, \mathcal{E})$ to $M$ subgraphs $G_1, G_2, ..., G_M$, and each subgraph $G_i$ can be viewed as a mini-batch $\mathcal{B}_i$ during training. Note that the number of nodes in each mini-batch should be (roughly) the same. Considering all $M$ mini-batches, the overall minimization objective becomes

$$\arg\min_{\{\mathcal{B}_1, ..., \mathcal{B}_M\}} \sum_{\mathcal{B}_i \in \{\mathcal{B}_1, ..., \mathcal{B}_M\}} \sum_{u \in \mathcal{B}_i} \sum_{v \in \mathcal{N}(u) \setminus \mathcal{B}_i} \sum_\ell C_v^\ell s_v^\ell$$

$$\text{subject to} \quad \mathcal{V} = \mathcal{B}_1 \cup \mathcal{B}_2 \cup ... \cup \mathcal{B}_M$$

$$\mathcal{B}_i \cap \mathcal{B}_j = \varnothing \quad \text{for all} \quad i \neq j, 1 \leq i, j \leq M \tag{4}$$

**Equivalence to graph partitioning objective.** Note that for $u \in \mathcal{B}_i$, $v \in \mathcal{N}(u) \setminus \mathcal{B}_i$ is equivalent to $v \in \mathcal{B}_j$ such that $i \neq j$ and $(u, v) \in \mathcal{E}$. Therefore, the objective in Equation 4 is equivalent to

$$\arg\min_{\{\mathcal{B}_1, ..., \mathcal{B}_M\}} \sum_{u \in \mathcal{B}_i, v \in \mathcal{B}_j, i \neq j, (u,v) \in \mathcal{E}} \sum_\ell C_u^\ell s_u^\ell + C_v^\ell s_v^\ell$$

$$\text{subject to} \quad \mathcal{V} = \mathcal{B}_1 \cup \mathcal{B}_2 \cup ... \cup \mathcal{B}_M$$

$$\mathcal{B}_i \cap \mathcal{B}_j = \varnothing \quad \text{for all} \quad i \neq j, 1 \leq i, j, \leq M \tag{5}$$

Then our optimization objective becomes a graph partitioning problem where we want to minimize the edge weight $e_{uv} = \sum_\ell C_u^\ell s_u^\ell + C_v^\ell s_v^\ell$ for all inter-partition edges. Therefore, we can use graph partitioning algorithms to minimize our objective and generate subgraphs (mini-batches) while explicitly minimizing the approximation error of learned node embeddings.

**Staleness score-based subgraph sampling.** Our objective aligns with the Kernighan-Lin objective (Kernighan & Lin, 1970) for graph partitioning problem, where we aim to minimize the total edge weight $e_{uv} = \sum_\ell C_u^\ell s_u^\ell + C_v^\ell s_v^\ell$ for all inter-partition edges. Nevertheless, it is often impractical to use the exact $\sum_\ell C_u^\ell s_u^\ell + C_v^\ell s_v^\ell$ as the edge weight since the computations of $C_u^\ell, C_v^\ell$ involve a lot of path-dependent factors as shown in Appendix B. Alternatively, we drop them to simplify the computations, i.e., $e_{uv} = \sum_\ell s_u^\ell + s_v^\ell$. Meanwhile, multi-level approaches proposed by Karypis & Kumar (1997); Dhillon et al. (2007) have been widely employed to efficiently solve the graph partitioning tasks based on similar objectives. In this way, our S3 sampling works as follows. We first define the weight of each edge $(u, v)$ as the sum of the staleness scores of the source and target nodes. Then we apply multi-level graph partitioning to generate mini-batches. In this way, we can explicitly reduce the approximation error of learned node embeddings.

## 3.3 FAST REFINEMENT ALGORITHM

One main challenge in our S3 sampling method is that the edge weight $e_{uv} = \sum_\ell s_u^\ell + s_v^\ell$ evolves dynamically throughout the training process. This dynamism arises from the continual updates applied

---

**Algorithm 1** Training of GAS with staleness score-based subgraph sampling (S3 + GAS). Note that for S3 + FM and S3 + LMC, lines 9-13 are different.

---

1: **Input:** Graph $G^0 = G = (\mathcal{V}, \mathcal{E})$, node features X, number of mini-batches $M$, training epochs $T$, GNN with learnable parameters $\theta$, number of GNN layers $L$
2: Preprocess by partitioning the input graph $G^0$ into $M$ subgraphs $G_1^0, G_2^0, ..., G_M^0$
3: **for** $(t = 0; t < T; t++)$ **do**
4:     Random shuffle $G_1^t, G_2^t, ..., G_M^t$
5:     **for** $(i = 1; i \leq M; i++)$ **do**
6:         Subgraph $G_i^t$ as a mini-batch $\mathcal{B}$ for training
7:         Pull historical embeddings $\bar{h}_v^1, \bar{h}_v^2, ..., \bar{h}_v^{L-1}$ for out-of-batch nodes $v \in \{\mathcal{N}(u) \backslash \mathcal{B} | u \in \mathcal{B}\}$
8:         $\tilde{h}_v^0 = \bar{h}_v^0 = x_v$
9:         **for** $(\ell = 0, \ell < L, \ell++)$ **do**
10:             Forward propagation and obtain new features by
$$\tilde{h}_u^{\ell+1} = f_\theta^{\ell+1}(\tilde{h}_u^\ell, \{\tilde{h}_v^\ell\}_{v \in \mathcal{N}(u) \cap \mathcal{B}} \cup \{\bar{h}_v^\ell\}_{v \in \mathcal{N}(u) \backslash \mathcal{B}})$$
11:             Push features $\tilde{h}_u^{\ell+1}$ for all in-batch nodes $u \in \mathcal{B}$ into historical embeddings
12:         **end for**
13:         Compute gradient and update model parameters $\theta$
14:     **end for**
15:     **for** $(\ell = 1, \ell < L, \ell++)$ **do**
16:         Forward propagation to compute full-neighborhood propagation embedding $h_v^\ell$
17:         Compute the staleness score $s_v^\ell = \|h_v^\ell - \bar{h}_v^\ell\|$ for each node $v$
18:     **end for**
19:     **if** Re-sampling (considering re-sampling scheduler as discussed in Section 3.2 and 4.2) **then**
20:         Define edge weight $e_{uv} = \sum_{\ell=1}^{L-1} s_u^\ell + s_v^\ell$
21:         Refinement from $G_1^t, G_2^t, ..., G_M^t$ to $G_1^{t+1}, G_2^{t+1}, ..., G_M^{t+1}$ by local refinement heuristics
22:     **else**
23:         Set $G_1^{t+1}, G_2^{t+1}, ..., G_M^{t+1}$ as $G_1^t, G_2^t, ..., G_M^t$
24:     **end if**
25: **end for**

---

to both historical embeddings and learnable parameters during training. Consequently, our problem transcends the traditional realm of graph partitioning, evolving into the domain of partitioning for dynamic graphs. To deal with the dynamic changes of staleness scores and improve the efficiency of graph partitioning, we design a fast algorithm to avoid re-partitioning the graph from scratch. In addition, we carefully design the re-sampling scheduler (frequency) based on empirical observations.

**Re-sampling scheduler.** To deal with the graph partitioning for dynamic graphs, one of the key factors is the partitioning frequency (scheduler). This hyperparameter plays a pivotal role in dictating when re-partitioning should be initiated, holding significant implications for the overall time complexity of the partitioning process. Practically, we find that conducting re-partitioning after a fixed number of epochs (*e.g.* 20 epochs) consistently yields favorable results without imposing significant time overhead. Detailed empirical analysis on the frequency is included in Section 4.2.

**Efficient refinement.** In addition, instead of re-partitioning the graph from scratch, we use $k$-way Kernighan–Lin refinement algorithm (Karypis & Kumar, 1998b) to do refinement, which is much more efficient. A detailed sampling and training framework is provided in Algorithm 1. Specifically, at the $t$-th training epoch, instead of directly partitioning the graph $G^t$ into $G_1^t, G_2^t, ..., G_M^t$, we perform refinement on the partitioning result $G_1^{t-1}, G_2^{t-1}, ..., G_M^{t-1}$ at the previous epoch. The refinement is based on the gain, i.e., the reduction in the edge cuts, by moving nodes to other mini-batches. Formally, for a node $u \in \mathcal{B}_i$, the potential gain of moving it from subgraph $G_i$ to $G_j$ is

$$g(u)_j = EW(u)_j - IW(u)$$
$$= \sum_{v \in \mathcal{N}(u) \cap \mathcal{B}_j} e_{uv} - \sum_{v \in \mathcal{N}(u) \cap \mathcal{B}_i} e_{uv}. \tag{6}$$

Here $EW(u)_j$ is called the external weight of node $u$ to subgraph $G_j$, and $IW(u)$ is the internal weight of node $u$. $g(u)_j$ is the reduction in the edge cuts by moving node $u$ from subgraph $G_i$ to $G_j$.

Table 1: Comparison between our sampling method and other baseline methods. Results for baseline methods are taken from Fey et al. (2021) and Shi et al. (2023). We apply our staleness score-based sampling method to the three popular and powerful historical embedding-based methods, namely GAS (Fey et al., 2021), GraphFM (Yu et al., 2022), and LMC (Shi et al., 2023). In the comparison with GAS and GraphFM, we use a fixed random seed following their original papers, therefore, we do not report standard deviation. For a fair comparison with LMC, we report the mean and standard deviation over five random runs. The three different background colors, gray , pink , and yellow , correspond to the three baseline methods. – indicates there is no reported value or it is hard to reproduce the reported value. Red indicates that our sampling method can improve the corresponding baselines with the default sampling. The top performance scores are highlighted in **bold**.

| # Nodes | | 89K | 230K | 717K | 169K | 2.4M |
|---|---|---|---|---|---|---|
| # Edges | | 450K | 11.6M | 7.9M | 1.2M | 61.9M |
| Method | | Flickr | Reddit | Yelp | ogbn-arxiv | ogbn-products |
| VR-GCN | | 0.482 ± 0.003 | 0.964 ± 0.001 | 0.640 ± 0.002 | – | – |
| FastGCN | | 0.504 ± 0.001 | 0.924 ± 0.001 | 0.265 ± 0.053 | – | – |
| GraphSAINT | | 0.511 ± 0.001 | 0.966 ± 0.001 | 0.653 ± 0.003 | – | 0.791 ± 0.002 |
| Cluster-GCN | | 0.481 ± 0.005 | 0.954 ± 0.001 | 0.609 ± 0.005 | – | 0.790 ± 0.003 |
| SIGN | | 0.514 ± 0.001 | 0.968 ± 0.000 | 0.631 ± 0.003 | 0.720 ± 0.001 | 0.776 ± 0.001 |
| GraphSAGE | | 0.501 ± 0.013 | 0.953 ± 0.001 | 0.634 ± 0.006 | 0.715 ± 0.003 | 0.783 ± 0.002 |
| GCN | GAS | 0.5400 | 0.9545 | – | 0.7168 | 0.7666 |
| | S3 + GAS | 0.5490 | 0.9548 | – | 0.7207 | 0.7715 |
| GCNII | GAS | 0.5620 | 0.9677 | 0.6514 | 0.7300 | 0.7724 |
| | S3 + GAS | 0.5666 | 0.9688 | 0.6524 | **0.7330** | 0.7750 |
| PNA | GAS | 0.5667 | 0.9717 | 0.6440 | 0.7250 | 0.7991 |
| | S3 + GAS | 0.5729 | 0.9718 | 0.6444 | 0.7303 | **0.8069** |
| GCN | FM | 0.5446 | 0.9540 | – | 0.7181 | – |
| | S3 + FM | 0.5486 | 0.9520 | – | 0.7214 | – |
| GCNII | FM | 0.5631 | 0.9680 | **0.6529** | 0.7310 | 0.7742 |
| | S3 + FM | 0.5663 | 0.9693 | 0.6516 | **0.7330** | 0.7760 |
| PNA | FM | 0.5710 | 0.9712 | 0.6450 | 0.7290 | 0.8047 |
| | S3 + FM | **0.5738** | **0.9719** | 0.6434 | 0.7292 | 0.8024 |
| GCN | LMC | 0.5380 ± 0.0014 | 0.9544 ± 0.0002 | – | 0.7144 ± 0.0023 | – |
| | S3 + LMC | 0.5407 ± 0.0008 | 0.9548 ± 0.0001 | – | 0.7210 ± 0.0022 | – |
| GCNII | LMC | 0.5536 ± 0.0049 | 0.9688 ± 0.0003 | – | 0.7276 ± 0.0022 | – |
| | S3 + LMC | 0.5616 ± 0.0020 | 0.9693 ± 0.0002 | – | 0.7311 ± 0.0010 | – |

Therefore, node $u$ is moved to $G_k$ such that $k = \arg\max_j g(u)_j$. In addition to decreasing edge cuts, balancing the number of nodes in each subgraph is another constraint we need to consider during refinement. Following Karypis & Kumar (1998b), we add another condition for moving the nodes. That is, the node $u$ is allowed to move from from $G_i$ to $G_j$ if and only if

$$|\mathcal{B}_j| + 1 \leq N_{\max}, \text{ and } |\mathcal{B}_i| - 1 \geq N_{\min}. \tag{7}$$

Here $N_{\max}$ and $N_{\min}$ are the maximum and minimum number of nodes in all subgraphs. Practically, we set $N_{\max}$ as $1.03 \times N/M$, and $N_{\min}$ as $0.9 \times N/M$ following Karypis & Kumar (1998b). Here $N$ is the total number of nodes, and $M$ is the number of subgraphs (mini-batches). This constraint ensures that there is no subgraph with too many or too few nodes.

## 4 EXPERIMENTS

**Datasets.** Following previous studies, we evaluate our staleness score-based sampling method S3 on 5 large-scale datasets, including Flickr (Zeng et al., 2019), Reddit (Hamilton et al., 2017), Yelp (Zeng et al., 2019), ogbn-arxiv (Hu et al., 2021) and ogbn-products (Hu et al., 2021). Detailed descriptions of the datasets are provided in Table 6 in Appendix C.

**Baselines.** The baseline methods include VR-GCN (Chen et al., 2017), FastGCN (Chen et al., 2018), GraphSAINT (Zeng et al., 2019), Cluster-GCN (Chiang et al., 2019), SIGN (Frasca et al., 2020), GraphSAGE (Hamilton et al., 2017), GAS (Fey et al., 2021), GraphFM (Yu et al., 2022), and LMC (Shi et al., 2023), covering node-wise, layer-wise, and subgraph sampling methods, historical embedding-based methods, and pre-processing methods, as discussed in Section 2. Results for baseline methods are directly taken from their original papers.

**Software and hardware.** The implementation of our sampling method is based on PyTorch (Paszke et al., 2019), PyGAS (Fey et al., 2021), PyTorch Sparse (Fey & Lenssen, 2019), PyG (PyTorch Geometric) (Fey & Lenssen, 2019), and METIS (Karypis & Kumar, 1997). For a fair comparison with GAS, GraphFM, and LMC, we follow their official code and only replace their sampling method with ours. All the experiments are conducted on one Nvidia GeForce RTX 2080 GPU.

## 4.1 MAIN EMPIRICAL RESULTS

Since our sampling method is specially designed for historical embedding-based methods, we select three most recent and powerful historical embedding-based backbone methods (GAS (Fey et al., 2021), GraphFM (Yu et al., 2022), and LMC (Shi et al., 2023)) to show the improvement of our S3 sampling. The comparison with baseline methods is shown in Table 1. Specifically, the results of GAS, GraphFM, and LMC are directly taken from their original papers using their default subgraph sampling method. Table 1 shows that our new sampling method can improve the performance of all three historical embedding-based methods (GAS, GraphFM, and LMC) on almost all datasets. The consistent improvements indicate the great effectiveness and generalizability of our S3 sampling method. Note that GraphFM and LMC are designed to improve GAS from different aspects. Therefore, we show detailed comparisons with GraphFM and LMC in the following part.

**Comparison with GraphFM.** Compared to GAS, the main advantage of GraphFM is that it can alleviate the staleness of historical embeddings by updating historical embeddings with a feature momentum step, which requires an additional computational cost than GAS. In our method, we aim to minimize the staleness scores by a new sampling strategy. From the results in Figure 1 and Table 2, we can observe that our S3 + GAS can outperform GraphFM and perform similarly to S3 + GraphFM on both testing accuracy and staleness scores, showing that our new sampling strategy is a more effective way to alleviate the staleness of historical embeddings and improve model performance.

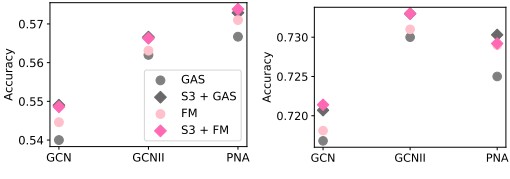

(a) Results on Flickr.  (b) Results on ogbn-arxiv.

Figure 1: Comparisons with GAS and GraphFM in terms of the testing results.

Table 2: Comparisons with GAS and GraphFM in terms of the staleness scores of historical embeddings. We use PNA as the GNN backbone. Hyperparameters follow the original setting in GAS and GraphFM.

| Datasets | Layer | GAS | S3 + GAS | FM | S3 + FM |
|---|---|---|---|---|---|
| Flickr | 1 | 2.2629 | 1.6924 | 1.775 | **1.6216** |
| | 2 | 2.6773 | 2.3254 | 2.411 | **2.2041** |
| | 3 | 3.8793 | **2.9235** | 3.2135 | 3.0828 |
| ogbn-arxiv | 1 | 3.0572 | **2.3210** | 2.6414 | 2.3482 |
| | 2 | 4.1617 | 3.874 | 3.9213 | **3.7477** |

**Comparison with LMC.** Compared to GAS, the main advantage of LMC is that it retrieves the discarded messages in backward passes, leading to accurate mini-batch gradients and thus accelerating convergence. Testing results in Table 1 and efficiency analysis in Table 3 show that our new sampling method can improve the performance of LMC without harming its efficiency and convergence.

Table 3: Comparison with LMC in terms of the number of epochs and the runtime.

| | | Flickr & GCN | Flickr & GCNII | ogbn-arxiv & GCN | ogbn-arxiv & GCNII |
|---|---|---|---|---|---|
| Epochs | LMC | 334.2 | 356 | 124.4 | 197.4 |
| | S3 + LMC | 362 | **211.4** | 175.4 | **180** |
| Runtime (s) | LMC | 85 | 475 | 55 | 178 |
| | S3 + LMC | 103 | **304** | 87 | **175** |

## 4.2 SAMPLING SCHEDULER

In our method, we re-sample based on the dynamic graph where the edge weights change during training. Similar to the learning rate scheduler, the re-sampling scheduler is an important hyperparameter governing our sampling steps. This critical hyperparameter exerts a substantial influence on both the testing performance and the overall efficiency of our method. We present an in-depth analysis of the outcomes obtained with various re-sampling schedulers. As shown in Table 4, re-sampling once is already better than not re-sampling. Re-sampling every fixed number of epochs (*e.g.* 20 epochs) consistently yields favorable results without imposing significant time overhead. The re-sampling times in Table 4 also demonstrate the efficiency of our refinement algorithm.

Table 4: Results for different sampling schedulers, and runtime for training and re-sampling. We use PNA as the GNN backbone. The two values in the last column correspond to the re-sampling time from scratch and the re-sampling time of our fast refinement algorithm, respectively.

| S3 + GAS | No | Resampling | Resample every a fixed number of epochs | | | | | Time per | Time per |
|---|---|---|---|---|---|---|---|---|---|
| | resampling | once | 80 | 40 | 20 | 8 | 1 | training epoch | resampling |
| Flickr | 0.5667 | 0.5683 | **0.5729** | 0.5711 | 0.5692 | 0.5715 | 0.5703 | 1s | 1s / 1s |
| ogbn-arxiv | 0.7250 | 0.7278 | 0.7291 | 0.7300 | **0.7303** | 0.7294 | 0.7292 | 1s | 3s / 2s |
| ogbn-products | 0.7991 | 0.8001 | 0.8030 | **0.8069** | 0.8035 | – | – | 40s | 120s / 48s |

### 4.3 ABLATION STUDY

In our sampling method, we first define the edge weight as the sum of the staleness scores of the two nodes. Subsequently, we partition this weighted graph into $M$ subgraphs, corresponding to $M$ mini-batches, by minimizing the total edge weights of inter-partition edges. This definition of edge weight and the associated minimization objective draw inspiration from our theoretical analysis, which aims to minimize the approximation error arising from the use of historical embeddings. This minimization is accomplished by reducing the sum of staleness scores for out-of-batch nodes. In this section, we underscore the significance of our defined edge weights and minimization objective by comparing it with different baseline approaches. The comparison results are detailed in Table 5.

Specifically, the "random node sampler" randomly partitions the input graph, whereas the other three methods partition the input graph by minimizing inter-partition edges. "No edge weight" implies that we disregard the staleness scores during

Table 5: Comparison of four sampling strategies. All results are based on the GAS baseline.

| Method | Flickr & PNA | ogbn-arxiv & PNA |
|---|---|---|
| Random node sampling | 0.5521 | 0.7104 |
| No edge weight | 0.5667 | 0.7250 |
| Random edge weight | 0.5671 | 0.7228 |
| Staleness score-based edge weight (ours) | **0.5729** | **0.7303** |

sampling, employing a graph partitioning algorithm solely to generate mini-batches. This is exactly the sampling method used in Cluster-GCN, GAS, GraphFM, and LMC. "Random edge weight" assigns random values to the edge weights. Meanwhile, "staleness score-based edge weight" is the approach introduced in this paper. Table 5 demonstrates that our solution surpasses all the variants, highlighting the effectiveness of our method with its optimal optimization objective.

In addition, as discussed in Section 3.2, the edge weight $e_{uv} = \sum_{\ell=1}^{L-1} s_u^\ell + s_v^\ell$ is defined as the sum of the staleness scores of both source and target nodes over all layers. But practically, we find that it is sufficient to only use the staleness scores at layer $L-1$ and set $e_{uv} = s_u^{L-1} + s_v^{L-1}$ to achieve similar results. Therefore, in our final experiments, we only use the staleness score of layer $L-1$ in our S3 sampling method. We believe this is because the staleness score increases monotonically across the layers due to error accumulation, as shown in Figure 2. Therefore, only using the staleness scores at layer $L-1$ can also minimize our optimization objective. Note that

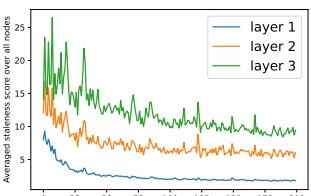

Figure 2: Staleness scores at different layers (ogbn-arxiv& GCNII).

we also normalize the staleness scores before performing our S3 sampling due to the presence of extreme values. Detailed discussion about the normalization is provided in Appendix C.

## 5 CONCLUSION

We focus on the task of training GNNs on large-scale graphs, where the main challenge is the neighbor explosion problem. Many of the existing methods use historical embeddings to solve this problem. In this paper, we first analyze the approximation error caused by using historical embeddings and show that the approximation error can be minimized by minimizing the staleness of the historical embeddings. We then design a staleness score-based subgraph sampling method to further benefit these historical embedding-based methods. Experimental results show that our sampling method can further improve historical embedding-based methods and set new state-of-the-art on various datasets.

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

# Appendix

## A  RELATED WORK

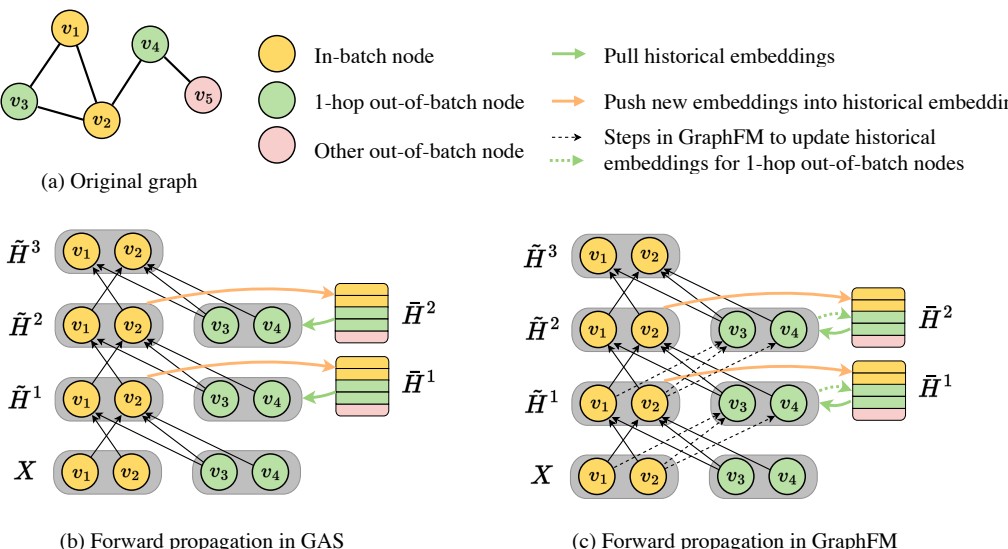

(a) Original graph

(b) Forward propagation in GAS

(c) Forward propagation in GraphFM

Figure 3: Illustrations for GAS and GraphFM, two historical embedding-based methods.

## B  PROOF OF THEOREM 1

*Proof.* **Path-based view of GNNs.** We can view a graph neural network with ReLUs as a directed acyclic computational graph and express the $i$-th output logit of node $u$ via paths through this graph (Gasteiger et al., 2022) as

$$h_{u,i}^L = C \sum_{v \in \mathcal{N}_{all}^L(u)} \sum_{p=1}^{\psi} \sum_{q=1}^{\phi} z_{v,p,i,q} x_{v,p,i,q} \prod_{\ell=1}^{L} a_{v,p}^\ell w_{i,q}^\ell,$$

where $C$ is a constant related to the size of the network (Choromanska et al., 2015), $\mathcal{N}_{all}^L(u)$ includes all nodes within $L$-hop of node $u$, $\psi$ is the total number of graph-based paths, $\phi$ is the total number of paths in learnable weights, $z_{v,p,i,q} \in \{0, 1\}$ denotes whether the path is active or inactive when any ReLU is deactivated, $x_{v,p,i,q}$ is the input feature used in the path, $a_{v,p}^\ell$ denotes the graph-dependent but feature-independent aggregation weight, and $w_{i,q}^\ell$ represents the used entry of the weight matrix $W_\ell$ at layer $\ell$.

**Aggregated embedding $\tilde{h}_u^L$ by using historical embeddings.** In the historical embedding based methods (Fey et al., 2021), the aggregated feature $\tilde{h}_u^L$ of node $u$ is based on the input features $x_v$ of in-batch nodes $\mathcal{N}_{in}^L(u)$ within $L$-hop of node $u$, input features $x_v$ of 1-hop out-of-batch nodes $\mathcal{N}_{out}^1(u)$, and the historical embeddings $\bar{h}_v$ of 1-hop out-of-batch nodes $\mathcal{N}_{out}^1(u)$, denoted as

$$\tilde{h}_{u,i}^L = \quad C \sum_{v \in \mathcal{N}_{in}^L(u)} \sum_{p=1}^{\psi_0^{in}} \sum_{q=1}^{\phi_0^{in}} z_{v,p,i,q} x_{v,p,i,q} \prod_{\ell=1}^{L} a_{v,p}^\ell w_{i,q}^\ell$$

$$+ C \sum_{v \in \mathcal{N}_{out}^1(u)} \sum_{p=1}^{\psi_{L-1}^{out}} \sum_{q=1}^{\phi_{L-1}^{out}} z_{v,p,i,q} \bar{h}_{v,p,i,q}^{L-1} a_{v,p}^L w_{i,q}^L$$

$$+ C \sum_{v \in \mathcal{N}_{out}^1(u)} \sum_{p=1}^{\psi_{L-2}^{out}} \sum_{q=1}^{\phi_{L-2}^{out}} z_{v,p,i,q} \bar{h}_{v,p,i,q}^{L-2} \prod_{\ell=L-1}^{L} a_{v,p}^\ell w_{i,q}^\ell$$

$$+\ldots$$

$$+C\sum_{v\in\mathcal{N}_{out}^{1}(u)}\sum_{p=1}^{\psi_{1}^{out}}\sum_{q=1}^{\phi_{1}^{out}}z_{v,p,i,q}\bar{h}_{v,p,i,q}^{1}\prod_{\ell=2}^{L}a_{v,p}^{\ell}w_{i,q}^{\ell}$$

$$+C\sum_{v\in\mathcal{N}_{out}^{1}(u)}\sum_{p=1}^{\psi_{0}^{out}}\sum_{q=1}^{\phi_{0}^{out}}z_{v,p,i,q}x_{v,p,i,q}\prod_{\ell=1}^{L}a_{v,p}^{\ell}w_{i,q}^{\ell}.$$

Note that $\mathcal{N}_{out}^{1}(u)$ is equivalent to $\mathcal{N}(u)\backslash\mathcal{B}$ in the main text. In the appendix, we use $\mathcal{N}_{out}^{1}(u)$ to simplify the notation.

**Full-neighborhood propagation embedding $h_{u}^{L}$.** Based on the path-based view of GNNs, the full-neighborhood propagation embedding $h_{u}^{L}$ can be formulated as

$$h_{u,i}^{L} = C\sum_{v\in\mathcal{N}_{all}^{L}(u)}\sum_{p=1}^{\psi}\sum_{q=1}^{\phi}z_{v,p,i,q}x_{v,p,i,q}\prod_{\ell=1}^{L}a_{v,p}^{\ell}w_{i,q}^{\ell}$$

$$= C\sum_{v\in\mathcal{N}_{in}^{1}(u)}\sum_{p=1}^{\psi_{L-1}^{in}}\sum_{q=1}^{\phi_{L-1}^{in}}z_{v,p,i,q}h_{v,p,i,q}^{L-1}a_{v,p}^{L}w_{i,q}^{L}$$

$$+C\sum_{v\in\mathcal{N}_{out}^{1}(u)}\sum_{p=1}^{\psi_{L-1}^{out}}\sum_{q=1}^{\phi_{L-1}^{out}}z_{v,p,i,q}h_{v,p,i,q}^{L-1}a_{v,p}^{L}w_{i,q}^{L}$$

$$= C\sum_{v\in\mathcal{N}_{in}^{2}(u)}\sum_{p=1}^{\psi_{L-2}^{in}}\sum_{q=1}^{\phi_{L-2}^{in}}z_{v,p,i,q}h_{v,p,i,q}^{L-2}\prod_{\ell=L-1}^{L}a_{v,p}^{\ell}w_{i,q}^{\ell}$$

$$+C\sum_{v\in\mathcal{N}_{out}^{1}(u)}\sum_{p=1}^{\psi_{L-2}^{out}}\sum_{q=1}^{\phi_{L-2}^{out}}z_{v,p,i,q}h_{v,p,i,q}^{L-2}\prod_{\ell=L-1}^{L}a_{v,p}^{\ell}w_{i,q}^{\ell}$$

$$+C\sum_{v\in\mathcal{N}_{out}^{1}(u)}\sum_{p=1}^{\psi_{L-1}^{out}}\sum_{q=1}^{\phi_{L-1}^{out}}z_{v,p,i,q}h_{v,p,i,q}^{L-1}a_{v,p}^{L}w_{i,q}^{L}$$

$$= C\sum_{v\in\mathcal{N}_{in}^{L}(u)}\sum_{p=1}^{\psi_{0}^{in}}\sum_{q=1}^{\phi_{0}^{in}}z_{v,p,i,q}x_{v,p,i,q}\prod_{\ell=1}^{L}a_{v,p}^{\ell}w_{i,q}^{\ell}$$

$$+C\sum_{v\in\mathcal{N}_{out}^{1}(u)}\sum_{p=1}^{\psi_{0}^{out}}\sum_{q=1}^{\phi_{0}^{out}}z_{v,p,i,q}x_{v,p,i,q}\prod_{\ell=1}^{L}a_{v,p}^{\ell}w_{i,q}^{\ell}$$

$$+C\sum_{v\in\mathcal{N}_{out}^{1}(u)}\sum_{p=1}^{\psi_{1}^{out}}\sum_{q=1}^{\phi_{1}^{out}}z_{v,p,i,q}h_{v,p,i,q}^{1}\prod_{\ell=2}^{L}a_{v,p}^{\ell}w_{i,q}^{\ell}$$

$$+\ldots$$

$$+C\sum_{v\in\mathcal{N}_{out}^{1}(u)}\sum_{p=1}^{\psi_{L-2}^{out}}\sum_{q=1}^{\phi_{L-2}^{out}}z_{v,p,i,q}h_{v,p,i,q}^{L-2}\prod_{\ell=L-1}^{L}a_{v,p}^{\ell}w_{i,q}^{\ell}$$

$$+C\sum_{v\in\mathcal{N}_{out}^{1}(u)}\sum_{p=1}^{\psi_{L-1}^{out}}\sum_{q=1}^{\phi_{L-1}^{out}}z_{v,p,i,q}h_{v,p,i,q}^{L-1}a_{v,p}^{L}w_{i,q}^{L}.$$

**Approximation error.** Then the difference between the full-neighborhood propagation embedding $h_{u,i}^L$ and the actual aggregated embedding $\tilde{h}_{u,i}^L$ is

$$
h_{u,i}^L - \tilde{h}_{u,i}^L = \quad C \sum_{v \in \mathcal{N}_{out}^1(u)} \sum_{p=1}^{\psi_{L-1}^{out}} \sum_{q=1}^{\phi_{L-1}^{out}} z_{v,p,i,q}(h_{v,p,i,q}^{L-1} - \bar{h}_{v,p,i,q}^{L-1}) a_{v,p}^L w_{i,q}^L
$$

$$
+ C \sum_{v \in \mathcal{N}_{out}^1(u)} \sum_{p=1}^{\psi_{L-2}^{out}} \sum_{q=1}^{\phi_{L-2}^{out}} z_{v,p,i,q}(h_{v,p,i,q}^{L-2} - \bar{h}_{v,p,i,q}^{L-2}) \prod_{\ell=L-1}^{L} a_{v,p}^\ell w_{i,q}^\ell
$$

$$
+ \dots
$$

$$
+ C \sum_{v \in \mathcal{N}_{out}^1(u)} \sum_{p=1}^{\psi_1^{out}} \sum_{q=1}^{\phi_1^{out}} z_{v,p,i,q}(h_{v,p,i,q}^1 - \bar{h}_{v,p,i,q}^1) \prod_{\ell=2}^{L} a_{v,p}^\ell w_{i,q}^\ell.
$$

Then squaring both sides of the equation, we have

$$
(h_{u,i}^L - \tilde{h}_{u,i}^L)^2 \leq \quad C^2 \sum_{v \in \mathcal{N}_{out}^1(u)} \sum_{p=1}^{\psi_{L-1}^{out}} \sum_{q=1}^{\phi_{L-1}^{out}} z_{v,p,i,q}(h_{v,p,i,q}^{L-1} - \bar{h}_{v,p,i,q}^{L-1})^2 (a_{v,p}^L w_{i,q}^L)^2
$$

$$
+ C^2 \sum_{v \in \mathcal{N}_{out}^1(u)} \sum_{p=1}^{\psi_{L-1}^{out}} \sum_{q=1}^{\phi_{L-2}^{out}} z_{v,p,i,q}(h_{v,p,i,q}^{L-2} - \bar{h}_{v,p,i,q}^{L-2})^2 \prod_{\ell=L-1}^{L} (a_{v,p}^\ell w_{i,q}^\ell)^2
$$

$$
+ \dots
$$

$$
+ C^2 \sum_{v \in \mathcal{N}_{out}^1(u)} \sum_{p=1}^{\psi_1^{out}} \sum_{q=1}^{\phi_1^{out}} z_{v,p,i,q}(h_{v,p,i,q}^1 - \bar{h}_{v,p,i,q}^1)^2 \prod_{\ell=2}^{L}(a_{v,p}^\ell w_{i,q}^\ell)^2.
$$

Therefore, the approximation error can be formulated as

$$
\|h_u^L - \tilde{h}_u^L\|_2^2 = \quad \sum_i (h_{u,i}^L - \tilde{h}_{u,i}^L)^2
$$

$$
\leq \quad C^2 \sum_{v \in \mathcal{N}_{out}^1(u)} \sum_{p=1}^{\psi_{L-1}^{out}} \sum_{q=1}^{\phi_{L-1}^{out}} \sum_i z_{v,p,i,q}(h_{v,p,i,q}^{L-1} - \bar{h}_{v,p,i,q}^{L-1})^2 (a_{v,p}^L w_{i,q}^L)^2
$$

$$
+ C^2 \sum_{v \in \mathcal{N}_{out}^1(u)} \sum_{p=1}^{\psi_{L-1}^{out}} \sum_{q=1}^{\phi_{L-2}^{out}} \sum_i z_{v,p,i,q}(h_{v,p,i,q}^{L-2} - \bar{h}_{v,p,i,q}^{L-2})^2 \prod_{\ell=L-1}^{L} (a_{v,p}^\ell w_{i,q}^\ell)^2
$$

$$
+ \dots
$$

$$
+ C^2 \sum_{v \in \mathcal{N}_{out}^1(u)} \sum_{p=1}^{\psi_1^{out}} \sum_{q=1}^{\phi_1^{out}} \sum_i z_{v,p,i,q}(h_{v,p,i,q}^1 - \bar{h}_{v,p,i,q}^1)^2 \prod_{\ell=2}^{L}(a_{v,p}^\ell w_{i,q}^\ell)^2
$$

$$
\leq \quad \sum_{v \in \mathcal{N}_{out}^1(u)} C_v^{L-1} \|h_v^{L-1} - \bar{h}_v^{L-1}\|_2^2
$$

$$
+ \sum_{v \in \mathcal{N}_{out}^1(u)} C_v^{L-2} \|h_v^{L-2} - \bar{h}_v^{L-2}\|_2^2
$$

$$
+ \dots
$$

$$
+ \sum_{v \in \mathcal{N}_{out}^1(u)} C_v^1 \|h_v^1 - \bar{h}_v^1\|_2^2
$$

$$
= \quad \sum_{v \in \mathcal{N}_{out}^1(u)} \sum_{\ell=1}^{L-1} C_v^\ell \|h_v^\ell - \bar{h}_v^\ell\|_2^2.
$$

Here $C_v^\ell$ is a weight that depends on both the graph structure and model parameters. Finally, we have

$$\|h_u^L - \tilde{h}_u^L\| \le \sum_{\ell=1}^{L-1} C_v^\ell \|h_v^\ell - \bar{h}_v^\ell\|.$$

The approximation error, the error (Euclidean distance) between the full-neighborhood propagation embedding $h_u^L$ and the actual aggregated embedding $\tilde{h}_u^L$, can be upper bounded by $\sum_{\ell=1}^{L-1} C_v^\ell \|h_v^\ell - \bar{h}_v^\ell\|$. Therefore, we can minimize the approximation error by minimizing $\sum_{\ell=1}^{L-1} C_v^\ell \|h_v^\ell - \bar{h}_v^\ell\|$. $\quad\square$

## C    EXPERIMENTAL DETAILS

**Datasets.** Detailed descriptions of the datasets are provided in Table 6.

Table 6: Statistics of the datasets. Here "m" indicates the multi-label classification task, and "s" indicates the single-label classification task.

| Dataset | # of nodes | # of edges | Avg. degree | # of features | # of classes | Train/Val/Test |
|---|---|---|---|---|---|---|
| Flickr | 89,250 | 899,756 | 10.0813 | 500 | 7(s) | 0.500/0.250/0.250 |
| Reddit | 232,965 | 11,606,919 | 49.8226 | 602 | 41(s) | 0.660/0.100/0.240 |
| Yelp | 716,847 | 6,997,410 | 9.7614 | 300 | 50(m) | 0.750/0.150/0.100 |
| ogbn-arxiv | 169,343 | 1,166,243 | 6.8869 | 128 | 40(s) | 0.537/0.176/0.287 |
| ogbn-products | 2,449,029 | 61,859,140 | 25.2586 | 100 | 47(s) | 0.100/0.020/0.880 |

**Hyperparameters.** Optimal hyperparameters for S3 + GAS, S3 + FM, and S3 + LMC are provided in Table 7. Specifically, we provide hyperparameters for sampling (e.g. resampling frequency), GNN model (e.g. GNN layers, hidden channels), and training (e.g. learning rate, batch size). Note that we follow the original hyperparameters of GAS, FM, and LMC, and only tune the learning rate scheduler, number of epochs, and the resampling scheduler.

**Normalization of staleness scores.** During training, we observe some extreme staleness scores which can lead to poor performance. Therefore, we normalize the staleness scores before performing our S3 sampling. Specifically, for staleness scores larger than a fixed value $x$, we set them to $x$. We choose $x$ as the value that is greater than 90% of the staleness scores for all nodes.

Table 7: Optimal hyperparameters for S3 + GAS, S3 + FM, and S3 + LMC. Here # layers indicates the number of GNN layers, and LR indicates the initial learning rate. # parts is the number of subgraphs we generate from the given input graph, and batch size is the number of subgraphs we select to build a mini-batch. Note that we do resampling after every fixed number of epochs, and Resampling in the table indicates that fixed number of epochs.

| | | | # layers | Hidden dim | LR | Epoch | # parts | Batch size | Resampling |
|---|---|---|---|---|---|---|---|---|---|
| Flickr | GCN | S3 + GAS | 2 | 256 | 0.01 | 1000 | 24 | 12 | 80 |
| | | S3 + FM | 2 | 512 | 0.01 | 1000 | 24 | 12 | 80 |
| | | S3 + LMC | 2 | 256 | 0.01 | 400 | 24 | 12 | 80 |
| | GCNII | S3 + GAS | 8 | 256 | 0.01 | 1000 | 24 | 12 | 80 |
| | | S3 + FM | 8 | 256 | 0.01 | 1000 | 24 | 12 | 80 |
| | | S3 + LMC | 8 | 256 | 0.01 | 400 | 24 | 12 | 20 |
| | PNA | S3 + GAS | 4 | 64 | 0.005 | 1000 | 24 | 12 | 80 |
| | | S3 + FM | 4 | 64 | 0.005 | 1000 | 24 | 12 | 20 |
| Reddit | GCN | S3 + GAS | 2 | 256 | 0.01 | 400 | 200 | 100 | 20 |
| | | S3 + FM | 2 | 256 | 0.05 | 400 | 200 | 100 | 40 |
| | | S3 + LMC | 2 | 256 | 0.01 | 400 | 200 | 100 | 80 |
| | GCNII | S3 + GAS | 4 | 256 | 0.01 | 400 | 200 | 100 | 40 |
| | | S3 + FM | 4 | 256 | 0.01 | 400 | 200 | 100 | 20 |
| | | S3 + LMC | 4 | 256 | 0.01 | 400 | 200 | 100 | 40 |
| | PNA_jk | S3 + GAS | 3 | 128 | 0.005 | 400 | 200 | 100 | 40 |
| | | S3 + FM | 3 | 128 | 0.005 | 400 | 200 | 100 | 80 |
| Yelp | GCNII | S3 + GAS | 2 | 512 | 0.01 | 500 | 40 | 5 | 20 |
| | | S3 + FM | 2 | 512 | 0.01 | 500 | 40 | 5 | 10 |
| | PNA | S3 + GAS | 3 | 512 | 0.005 | 400 | 40 | 5 | 20 |
| | | S3 + FM | 3 | 512 | 0.005 | 400 | 40 | 5 | 20 |
| ogbn-arxiv | GCN | S3 + GAS | 3 | 256 | 0.01 | 300 | 80 | 40 | 20 |
| | | S3 + FM | 3 | 256 | 0.01 | 400 | 80 | 40 | 20 |
| | | S3 + LMC | 3 | 256 | 0.01 | 300 | 80 | 40 | 10 |
| | GCNII | S3 + GAS | 4 | 256 | 0.01 | 500 | 40 | 20 | 20 |
| | | S3 + FM | 4 | 256 | 0.01 | 500 | 40 | 20 | 40 |
| | | S3 + LMC | 4 | 256 | 0.01 | 500 | 40 | 20 | 40 |
| | PNA | S3 + GAS | 3 | 256 | 0.005 | 500 | 40 | 20 | 20 |
| | | S3 + FM | 3 | 256 | 0.005 | 300 | 40 | 20 | 40 |
| ogbn-products | GCN | S3 + GAS | 2 | 256 | 0.005 | 300 | 7 | 1 | 20 |
| | GCNII | S3 + GAS | 5 | 128 | 0.001 | 240 | 150 | 1 | 40 |
| | | S3 + FM | 5 | 128 | 0.005 | 300 | 150 | 1 | 40 |
| | PNA | S3 + GAS | 3 | 256 | 0.001 | 150 | 150 | 1 | 40 |
| | | S3 + FM | 3 | 256 | 0.001 | 150 | 150 | 1 | 20 |

