# OpenReview forum: "Staleness-based subgraph sampling for large-scale GNNs training"
_ICLR.cc/2024/Conference — Submitted to ICLR 2024_

### Official Review · Reviewer_XXgf · 2023-10-23

**Soundness:** 2 fair
**Presentation:** 3 good
**Contribution:** 1 poor
**Rating:** 3
**Confidence:** 4

**Summary:**

This paper proposes S3, a staleness aware subgraph sampling method for GNN training on large graphs. It first shows that the final approximation error of node embedding is related to the error caused by stale embedding. Then, it uses the stale error of two nodes as the weight of an edge and conducts graph partitioning by minimizing the weight of the cross-partition edges. To reduce the overhead of graph partition, it proposes to adjust partition adjustment instead of running from scratch.

**Strengths:**

1.	The paper is well-written and easy to follow.
2.	The proposed method makes sense.

**Weaknesses:**

1.	Theorem 1 is wrong. First, Appendix B proves that the latter part is an upper bound of the former part. There is no guarantee that minimizing an upper bound for an expression (i.e., the latter part) will actually minimize the expression (the former part). Thus, Theorem 1 should be stated in much weaker form. Second, on the bottom of Page 16, the expressions use the squared approximation error, but on the top of page 17, it becomes the approximation error (without square for the Euclidean norm). If this is not a mistake, it should be made clear how the derivation works.
2.	The performance gain is very limited, and, in most cases, the improvements in accuracy happen only three digits after zero. To make a practical impact, much larger improvements are required.
3.	Experiment needs to be improved. (1) In Table 1, the results of many cases are missing, and the explanation is that these results are not reported in their original paper or difficult to reproduce. It should be made clear which results are not reported and which results are difficult to reproduce; for these difficult to reproduce, pls specify why; for these not reported, pls try the best to run the experiments, if you cannot, pls explain the specific reasons; at least provide the results of your method in these cases. (2) Although the paper claims to support very large datasets, the datasets used are actually quite small. Some large datasets are well-known for GNN training, e.g., Papers-100M, MAG240, and IGB. Pls consider using these datasets for the experiment. But I wonder if the many nodes in these graphs will make graph partitioning expensive. (3) To validate the necessity of partitioning adjustment, the authors make check the difference of a graph partition before and after the adjustment. This can be measured by the portion of nodes that change their partition. (4) The influence of the number of partitions.

**Questions:**

My primary concern of this paper is the limited accuracy gain, which makes the practical impact marginal.

---

> ### Comment · Reviewer_XXgf · 2023-11-22
> **After response**
>
> I have read the comments of othe reviewers and decided to keep the original rating.

---

### Official Review · Reviewer_BmCi · 2023-10-31

**Soundness:** 3 good
**Presentation:** 3 good
**Contribution:** 2 fair
**Rating:** 5
**Confidence:** 3

**Summary:**

This paper proposed a subgraph sampling method that can benefit historical embedding-based large-scale graph training method. This seems to be the first work considering what kind of subgraph sampling is better for historical embeddings. The authors design a staleness score for subgraph sampling and provide a simple heuristic algorithm for constructing mini-batches. Experimental results show that S3 improves the performance of three historical embedding-based methods.

**Strengths:**

s1.The motivation of the article is  reasonable, I agree with the authors' viewpoint that using simple methods like METIS to construct mini-batches is not suitable for historical embeddings.
s2.The analysis part about S3 sampling is reasonable, I believe it's a simple and effective method that can be applied to most historical embedding methods.
s3.From the experimental results, re-sampling does not require too much time, even without re-sampling, the performance of S3 sampling is acceptable.
s4.The ablation experiments have proven the effectiveness of S3 sampling.

**Weaknesses:**

w1.The improvement in accuracy of S3 sampling on some datasets, such as Reddit, is very small. This diminishes the necessity of S3 sampling.
w2.Section 3.3 about refinement is too briefly written, and its readability needs to be improved. I hope the authors can provide a more detailed explanation.

**Questions:**

No

---

### Official Review · Reviewer_No3a · 2023-11-01

**Soundness:** 2 fair
**Presentation:** 3 good
**Contribution:** 2 fair
**Rating:** 5
**Confidence:** 3

**Summary:**

This paper proposes a novel Staleness score-based Subgraph Sampling method to benefit those historical embedding-based methods. The proposed method defines the edge weight as the sum of the staleness scores of the source and target nodes, and partitions the graph into mini-batches. Furthermore, to deal with the dynamic changes of staleness scores during training, the authors design a fast algorithm to generate mini-batch via a local refinement heuristic. Experiments demonstrated the efficiency of the proposed S3 method.

**Strengths:**

Strengths:
a)	The motivations of this work are clear.
b)	This paper has sufficient experiments, and the dataset used is relatively common.

**Weaknesses:**

Weaknesses:
a)	The backbone method GAS、GraphFM、LMC are all works done before, and the graph partitioning method is much like Minimum Cut algorithm which utilizes the weight of edges. And the refinement algorithm is also based on Kernighan-Lin algorithm.
b)	In Algorithm1, it computes full-neighborhood forward propagation, and calculates the staleness score for each node v in the graph. It will case exponential explosion problem as the author mentioned in the background section. So the paper is not technically sound.
c)	The improvement in model accuracy is limited.

**Questions:**

1. Are the hyperparameters setting optimal, and have you tried other hyperparameters settings?
2. Have you tested the time consumption and the memory consumption on your method? Is the computation time to compute full-neighborhood forward propagation and calculate the staleness score for each node v affordable? And will it occupy too much memory?
3. Have you compared the model accuracy your method get with the full-neighborhood model accuracy? Only applying full-neighborhood forward propagation will result in how much difference from the full-neighborhood model?
4. Is there a huge gap between the embedding h_v^l calculated by the model parameters θ updated from the staleness-based method and the node embedding  h_v^l calculated by the full-neighborhood model?

---

### Official Review · Reviewer_Ummh · 2023-11-01

**Soundness:** 2 fair
**Presentation:** 3 good
**Contribution:** 2 fair
**Rating:** 3
**Confidence:** 4

**Summary:**

This paper presents S3, a sampling method for reducing approximation error incurred by stale embeddings. To do so, this work proposes re-partition paradigm so that a pair of neighbors are likely to be separated if their approximation errors are small. Experiments show that S3 can improve the accuracy of GAS, GraphFM, and LMC. S3 also theoretically proves that weighted aggregation can minimize the approximation error, but computing this set of weights is expensive.

**Strengths:**

1. The proposed approach is novel. Leveraging periodic graph partition to reduce staleness error is an interesting research direction.
2. Experimental results show that S3 consistently improves the accuracy of existing works.

**Weaknesses:**

1. This paper aims at improving the scalability of GNN training. However, if I understand the proposed technique correctly, the process of recomputing edge weight is not scalable. As shown in line 16 in Algorithm 1, $L$ rounds of full-graph aggregation are required for computing staleness score.

2. As the accurate aggregations are computed in line 16, why do we need to use historical embeddings after this step? In addition, LLCG [1] leverages periodical full-graph aggregation. I suspect that LLCG performs better than S3.

3. This paper aims at improving the scalability and efficiency of GNN training, but the largest dataset used in this work is ogbn-products. Evaluating the performance of S3 on ogbn-papers100M is highly appreciated especially as one recent work (see Table 3 in ReFresh [2]) shows that GraphFM suffers from poor accuracy on ogbn-papers100M.

4. The efficiency comparison is not comprehensive. The experiments for GAS and GraphFM are missing.

5. The convergence comparison is missing.



[1] Learn Locally, Correct Globally: A Distributed Algorithm for Training Graph Neural Networks

[2] ReFresh: Reducing Memory Access from Exploiting Stable Historical Embeddings for Graph Neural Network Training

**Questions:**

1. For the first weakness, how is this step (line 16 in Algorithm 1) implemented? What's the processing time and memory requirement for computing this step? The draft only reports the overhead of graph partition but I feel that $L$ rounds of full-graph aggregation is more time-consuming than graph partition.

2. If periodical full-graph aggregation is required, please compare S3 with LLCG as it requires similar resources.

3. What's the performance of S3 on ogbn-papers100M?

4. What's the efficiency of S3+GAS and S3+GraphFM? Why S3 can improve the efficiency of LMC for some cases in Table 3?

5. Please show the convergence comparison between X with S3+X where X is any of the baselines you choose.

6. What is the underlying assumption about the aggregation function made in Theorem 1? I think it cannot be applied to Max and Min aggregations which are adopted in PNA.

7. How to determine $C$ in equation 3? If the expression is complex, please provide some high-level explanations so that the readers can better understand this theorem.

8. This is one minor question. I feel that Table 2 is not informative enough. Could you please compare the trend of the staleness score? You may refer to Figure 5 in [1].

[1] PipeGCN: Efficient Full-Graph Training of Graph Convolutional Networks with Pipelined Feature Communication

---

### Author Response · Authors · 2023-11-23
**General response**

We thank all the reviewers for their valuable comments and appreciate that all the reviewers find that
- our presentation is good,
- the proposed method is well-motivated (novel, clear motivation, reasonable, makes sense), and
- the experimental results are good (consistently improves existing works, sufficient experiments and commonly used datasets, acceptable performance).

Here are some detailed responses, and we will add them in our next version. Thanks again for all the comments which are very helpful!

> Implementation details and complexity for full-neighbor aggregation (Line 16 in Algorithm 1)

  W1, W2, Q1, Q2 by `Reviewer Ummh`, W(b) by `Reviewer No3a`

  - Firstly, we want to emphasize that **the implementations of previous methods (GAS, FM, and LMC) already have this full-neighbor aggregation step**. And based on their implementation, **the main changes we added are calculating staleness score (error between two embeddings, easy to compute), and the new S3 sampling**. Therefore, we mainly care about the additional time cost for the S3 sampling and provide a comparison in Table 4.
  - Specifically, we (FM, LMC, and our S3) all follow the code of GAS. In GAS, there is a special design during the evaluation to do this full-neighbor aggregation, as shown [here](https://github.com/rusty1s/pyg_autoscale/blob/master/torch_geometric_autoscale/models/base.py#L200).
  - Practically, on the ogbn-products dataset with 2M nodes, GAS (PNA) + S3 takes 18s for one training epoch, 22s for the evaluation (full-neighbor aggregation), and 48s for one resampling.

> Experimental results: some missing parts in the table, the improvement over baseline methods

  W4, W5, Q4, Q5 by `Reviewer Ummh`, W(c) by `Reviewer No3a`, W1 by `Reviewer BmCi`, W2, W3(1) by `Reviewer XXgf`

  -  About the missing parts in Table 1:
     - For the comparison between GAS and GAS + S3, we cannot reproduce the results for Yelp & GCN + GAS (reported: 0.6294 vs reproduced: around 0.4). Therefore, we don't compare to GAS in this setting.
     - For the comparison between FM and FM + S3, FM doesn't provide results for Yelp & GCN + GAS. Therefore, we don't compare to FM in this setting. In addition, for ogbn-products & GCN + FM, we cannot reproduce the results (reported: 0.7688 vs reproduced: 0.7149).
     - For the comparison between LMC and LMC + S3, LMC doesn't provide results on Yelp and ogbn-products. Therefore, we don't compare to LMC on these two datasets.

  - Improvement over baseline methods
    - Firstly, as shown in Table 1, our new S3 sapling can **consistently outperform all baseline methods**, this demonstrates the great generalizability of our S3 sampling.
    - Specifically, compared to GAS, the main advantage of FM is that it can alleviate the staleness of historical embeddings by updating historical embeddings with a feature momentum step. **Figure 1 and Table 2 show that compared to FM, our S3 is a more effective way to alleviate the staleness of historical embeddings and improve model performance.**
    - Compared to GAS and FM, the main advantage of LMC is that it can accelerate convergence. **From Table 3, we show that our S3 can improve the performance of LMC without harming its efficiency and convergence.**

> Considering larger datasets

  W3, Q3 by `Reviewer Ummh`, W3(2) by `Reviewer XXgf`

  Thanks for the great suggestion!
  Although we have included the most commonly used datasets, we agree that testing our method on more large-scale and real-world datasets is more practically useful.

  Honestly, we were aware of the exciting results provided in ReFresh [1] on ogbn-papers 100M dataset, however, we didn't find the corresponding public code. Therefore, we didn't consider this dataset in our current version.

  We are working on the experiments on this data and will include it in our next version.

[1] ReFresh: Reducing Memory Access from Exploiting Stable Historical Embeddings for Graph Neural Network Training

---

### Meta-Review · Area_Chair_onGN · 2023-11-29

**Metareview:**

This paper introduces S3, a subgraph sampling method aimed at reducing approximation errors in large-scale graph training using historical embeddings. S3 employs a novel approach, the Staleness score-based Subgraph Sampling, where edge weights are determined by the sum of staleness scores for source and target nodes. The graph is partitioned into mini-batches using a re-partition paradigm, ensuring that pairs of neighbors with small approximation errors are likely to be separated. Additionally, the paper addresses dynamic staleness changes during training through a fast algorithm based on local refinement heuristics. Experimental results demonstrate the efficacy of S3, enhancing the performance of historical embedding-based methods such as GAS, GraphFM, and LMC. The paper also theoretically establishes that weighted aggregation minimizes approximation errors, albeit with computational expense.

The proposed method may be novel. However, there are some concerns raised by reviewers that are not well addressed. For example the question about Theorem 1 (Question 6 of Ummh). Thus, it needs further modification of the paper before publication. I encourage authors to revise the paper based on the reviewer's comments and resubmit it to a future venue.

**Justification For Why Not Higher Score:**

All reviewers agree to reject the paper and I also agree the decision.

**Justification For Why Not Lower Score:**

N/A

---

### Decision · Program_Chairs · 2024-01-16

Reject